# Human Milk Oligosaccharide 2′-Fucosyllactose Induces Neuroprotection from Intracerebral Hemorrhage Stroke

**DOI:** 10.3390/ijms22189881

**Published:** 2021-09-13

**Authors:** Tsai-Wei Hung, Kuo-Jen Wu, Yu-Syuan Wang, Eun-Kyung Bae, YoungHa Song, JongWon Yoon, Seong-Jin Yu

**Affiliations:** 1Center for Neuropsychiatric Research, National Health Research Institutes, Zhunan 35053, Taiwan; pilypily@nhri.edu.tw (T.-W.H.); kjwu@nhri.edu.tw (K.-J.W.); yswang@nhri.edu.tw (Y.-S.W.); baee@nhri.edu.tw (E.-K.B.); 2Advanced Protein Technologies Corporation, Suwon-si 16229, Korea; yhsong@aptech.biz (Y.S.); jongwon.yoon91@gmail.com (J.Y.)

**Keywords:** 2-fucosyllactose, intracerebral hemorrhage stroke, inflammation, endoplasmic reticulum

## Abstract

Intracerebral hemorrhage (ICH) occurs when brain blood vessels rupture, causing inflammation and cell death. 2-Fucosyllactose (2FL), a human milk oligosaccharide, has potent antiapoptotic and anti-inflammatory effects. The purpose of this study was to examine the protective effect of 2FL in cellular and rodent models of ICH. Hemin was added to a primary rat cortical neuronal and BV2 microglia coculture to simulate ICH in vitro. IBA1 and MAP2 immunoreactivities were used to determine inflammation and neuronal survival. Hemin significantly increased IBA1, while it reduced MAP2 immunoreactivity. 2FL significantly antagonized both responses. The protective effect of 2FL was next examined in a rat ICH model. Intracerebral administration of type VII collagenase reduced open-field locomotor activity. Early post-treatment with 2FL significantly improved locomotor activity. Brain tissues were collected for immunohistochemistry and qRT-PCR analysis. 2FL reduced IBA1 and CD4 immunoreactivity in the lesioned striatum. 2FL downregulated the expression of ER stress markers (PERK and CHOP), while it upregulated M2 macrophage markers (CD206 and TGFβ) in the lesioned brain. Taken together, our data support that 2FL has a neuroprotective effect against ICH through the inhibition of neuroinflammation and ER stress. 2FL may have clinical implications for the treatment of ICH.

## 1. Introduction

Intracerebral hemorrhage (ICH) is a major neurological disorder associated with a high risk of death in adults [1,2]. ICH constitutes 10–15% of all acute stroke cases [3]. A few epidemiological studies have supported gender disparities in patients with ICH [4] or cerebrovascular diseases [5]. A small portion of ICH patients develop hemorrhagic lacunar syndrome [6]. Current treatment for ICH focuses on controlling the bleeding, removing the blood clot, and relieving intracranial pressure caused by bleeding. There is no effective pharmacological therapy for ICH.

Inflammation plays a critical role in ICH-induced neurodegeneration. After the onset of ICH, blood cells enter the brain parenchyma, activate the microglia, and initiate the production of proinflammatory cytokines [7,8]. Hemin, a byproduct of hemoglobin, causes neuronal death and induces apoptosis and inflammation [9,10]. Several studies have supported the effectiveness of anti-inflammatory treatment in animal models of ICH [11,12]. We recently demonstrated that early post-treatment with a CXCR4 antagonist CX807 inhibited inflammation and improved behavioral function in experimental ICH rats [13]. These data suggest that anti-inflammatory therapy may reduce brain damage after ICH.

Human breastmilk contains nutrition and several critical bioactive ingredients [14], e.g., human milk oligosaccharides (HMOs) [15]. 2′-Fucosyllactose (2FL) is a major oligosaccharide in human milk. Oral delivery of 2FL improved memory and learning in rats [16,17]. In addition, 2FL suppressed lipopolysaccharide-induced CD14 expression and inflammation in human enterocytes [18]. We recently reported that 2FL mitigated neuroinflammation in a brain suffering ischemic stroke [19]. These data suggest that 2FL is a potent anti-inflammatory agent and may, thus, be useful for the treatment of ICH.

The purpose of this study was to examine the neural protective actions of 2FL in cellular and animal models of ICH. Hemin was used to simulate ICH in cell culture [20]. We demonstrated that 2FL antagonized hemin-mediated activation of microglia. Type VII collagenase was used to generate focal ICH in adult rats [13]. We found that 2FL significantly mitigated ICH-induced microglia activation, CD4^+^ lymphocyte infiltration, and ER stress in the lesioned brain and improved behavioral function. Our data support that 2FL has a neuroprotective effect against ICH.

## 2. Results

### 2.1. 2FL Reduced Hemin-Mediated Neurodegeneration in Primary Cortical Neurons and BV2 Microglia Coculture

The protective effect of 2FL was first examined in a cellular model of cerebral hemorrhage. Primary cortical neuronal and BV2 microglia cocultures were treated with hemin (10 μM) for 48 h (see timeline in Figure 1E). Hemin attenuated neuronal marker MAP2 (Figure 1A), while it increased microglia marker IBA1 (Figure 1B). These responses were both antagonized by 2FL (1 μM, Figure 1A,B). The immunoreactivities of MAP2 and IBA1 were further analyzed by measuring the pixel density of specific markers. As seen in Figure 1C,D, hemin significantly reduced MAP2 immunoreactivity (Map2-ir) and increased IBA1-ir (*p* < 0.001, one-way ANOVA, *n* = 6 in each group). These responses were significantly antagonized by 2FL (MAP2: *p* < 0.001, F_2,14_ = 115.863; IBA1: *p* < 0.001, F_2,15_ = 134.644, one-way ANOVA).

### 2.2. Post-Treatment with 2FL Improved Locomotor Activity in ICH Rats

The protective effect of 2FL was next examined in an animal model of ICH. A total of 14 adult male rats were stereotactically administered a low dose (0.5 units/μL × 1 μL) of type VII collagenase to induce cerebral hemorrhage in the right striatum. 2FL (400 mg/kg/d, i.p., *n* = 7) or vehicle (saline, i.p., *n* = 7) was given daily for 5 days, starting 1 day after stroke (please see the timeline in Figure 2G). Then, 2 h after the last 2FL or vehicle injection on day 5, animals were individually placed in infrared activity chambers for 30 min. Open-field locomotor behavior was examined for 30 min. We found that 2FL significantly increased vertical activity (VACTV, Figure 2A), total distance traveled (TOTDIST, Figure 2B), vertical time (VTIME, Figure 2C), horizontal activity (HACTV, Figure 2D), movement time (MOVTME, Figure 2E), and movement number (MOVNO, Figure 2F). Detailed statistics (two-way ANOVA + post hoc Newman–Keuls (NK) tests) are listed in Table 1.

### 2.3. Early Post-Treatment with 2FL Reduced Microglial Activation in the Peri-Lesioned Region

Eight ICH rats (2FL, *n* = 4; veh, *n* = 4) were euthanized and perfused for IBA1 immunohistochemistry after the behavioral test on day 5. Typical IBA1-ir is demonstrated in Figure 3. IBA1-ir was enhanced in the peri-lesioned area in striatum (Figure 3B), as compared with the corresponding sites in the contralateral non-lesioned striatum (Figure 3A) in stroke rats receiving veh. At a high magnification, de-ramified or ameboid microglial cells were found in the lesioned area (Figure 3B, insert), while resting microglia exhibiting a ramified morphology were found in the non-lesioned striatum (Figure 3A, insert). 2FL treatment reduced IBA1-ir and partially restored morphological ramification of the microglia (Figure 3C, insert). IBA1-ir was quantified and averaged across three consecutive brain sections with a visualized anterior commissure in all animals. 2FL significantly reduced IBA1-ir in the ICH brain (Figure 3D, *p* < 0.001, F_2,12_ = 50.357, one-way ANOVA).

### 2.4. 2FL Mitigated CD4^+^ Lymphocyte Infiltration into the Lesioned Brains

Administration of collagenase enhanced CD4^+^ lymphocyte infiltration into the lesioned striatum (Figure 4B). Almost no CD4 cells were found in the non-lesioned striatum (Figure 4A). CD4 cell infiltration into the lesioned striatum was mitigated by 2FL treatment (Figure 4C vs. Figure 4B; *p* < 0.001, F_2,6_ = 173.485, one-way ANOVA, Figure 4D).

### 2.5. 2FL Differentially Altered the Expression of M1 and M2 Inflammatory Markers in ICH Brains

Striatal tissues were collected from eight rats (veh, *n* = 4; 2FL, *n* = 4) on day 5. The expression of M1/M2 inflammatory markers was examined by qRT-PCR. ICH significantly increased the expression of M1 marker CD86 (Figure 5A, *p* = 0.005) and M2 markers CD206 and TGFβ (Figure 5B, C, *p* = 0.005). 2FL did not alter the expression of CD86 (Figure 5A, *p* = 0.356, two-way ANOVA). However, it significantly upregulated the expression of CD206 (Figure 5B) and TGFβ in the lesioned side (*p* < 0.05, NK test).

### 2.6. 2FL Altered the Expression of ER Stress and Apoptotic Markers in ICH Brains

The expression of ER stress and apoptotic markers was examined in eight rats (veh, *n* = 4; 2FL, *n* = 4). 2FL significantly reduced the expression of PERK (Figure 6A, *p* = 0.016), IRE1 (Figure 6B, *p* = 0.018), CHOP (Figure 6C, *p* = 0.032), and SigmaR1 (Figure 6D, *p* = 0.017) in the lesioned striatum. 2FL did not alter the expression of BIP (Figure 6E, *p* = 0.841) and ATF6 (Figure 6F, *p* = 0.403). In addition, the expression of caspase3 was significantly reduced by 2FL (Figure 6G, *p* = 0.029, *t*-test).

## 3. Discussion

In this study, 2FL reduced hemin-induced microglia activation and neurodegeneration in a primary cortical neuron and BV2 microglia coculture. 2FL improved locomotor activity in ICH rats. Using immunohistochemical and qPCR analysis, we demonstrated that 2FL inhibited microglia activation, CD4(+) lymphocyte infiltration, and the expression of inflammatory and ER stress markers in the ICH brain. The main finding of this study is that 2FL is neuroprotective against ICH injury.

ICH causes acute and irreversible brain damage. Following the acute insult, a series of cascade reactions turn on, which result in chronic secondary degeneration. Hemoglobin and its degradation products are closely associated with secondary injury. Hemin is cytotoxic and contributes to brain damage, accompanied by hemorrhagic stroke [21,22]. Excessive hemin catalyzes free-radical chain reactions [23] and facilitates apoptosis and mitochondrial fission [24]. Hemin also potentiates microglia activation and aggravates inflammatory injury after intracerebral hemorrhage [25,26]. In this study, hemin was used to simulate ICH in a neuron and BV2 microglia coculture. We demonstrated that hemin was neurotoxic and activated microglia. Treatment with 2FL reduced hemin-mediated IBA1 activation and restored MAP2 immunoreactivity. Our data suggest that 2FL is anti-inflammatory and neuroprotective in a cellular model of ICH.

Previously, we demonstrated that local collagenase infusion resulted in ICH and bradykinesia in rats [13]. In this study, a similar animal model was used to examine the protective effect of 2FL in vivo. We demonstrated that systemic application of 2FL for 5 days improved locomotor movements in ICH rats. As inflammation plays a critical role in the progression of ICH [7,8,27], we also found microglia activation in the ICH brain. 2FL reduced IBA1-ir and partially restored ramification of microglia in the lesioned ICH brain. Moreover, we found that 2FL upregulates the expression of M2 (anti-inflammatory) microglia/macrophage inflammatory makers in the ICH rat brain. These data suggest that 2FL suppressed ICH-mediated microglia activation in the lesioned brain.

ICH promotes the migration of peripheral immune cells, such as CD4^+^ and CD8^+^ T cells, to the lesioned brain [28,29]. We previously reported the expression of cytotoxic T-cell markers in the ICH brain [13]. In addition, the peak infiltration of CD4 T cells was between days 3 and 4 after ischemic injury in the mice [30]. In this study, we demonstrated that ICH increased CD4^+^ lymphocyte infiltration into the lesioned striatum on day 5 in rats; CD4 cell infiltration was significantly mitigated by 2FL. These data support the notion that 2FL inhibits T-cell migration from the periphery to the ICH brain.

Previous studies have indicated that 2FL has protective effects in the periphery. In human enterocytes, 2FL attenuated the induction of CD14 [18] and the expression of IL-8, IL-1b, and MIP-2 [31]. 2FL also attenuates the severity of necrotizing enterocolitis in mouse neonatal intestine [32]. In this study, we demonstrated that 2FL has anti-inflammatory and neuroprotective effects in the ICH brain. Other studies also support that 2FL improved learning and hippocampal long-term potentiation in rodents [17], as well as prevented cell death in the ischemic brain [19]. These data suggest that 2FL has multifaceted beneficial effects against CNS and peripheral inflammation and degeneration.

ICH can induce secondary neurodegeneration through ER stress [33]. For example, the PERK pathway was activated in the ICH brain, as evidenced by the upregulation of p-eIF2α and ATF4 [34]. The resultant ER stress further induced neuronal apoptosis and cell death. We also reported that the expression of PERK, IRE1, CHOP, SigmaR1, and caspase-3 was enhanced in the ICH brain. 2FL significantly inhibited these responses. The detailed mechanisms underlying the regulation of ER stress by 2FL warrant investigation.

There are a few limitations to this study. We did not use the size of the hematoma to evaluate the outcome after 2FL therapy. As indicated previously, quantification of hematoma area on histological slices is usually labor-intensive and sometime subjective [35]. A new approach was recently developed to allow accurate and efficient measurements of cerebral hematoma volume [35]. It will be of interest to determine the association of hematoma volume with improvements in locomotor activity after 2FL treatment using this new approach. Our study was conducted in cellular and animal models of ICH. Additional nonhuman primate studies and prospective randomized trials of 2FL treatment in human subjects are required before its clinical use.

Human milk is regarded as the best source of nutrition for newborns and developing infants. Beyond its nutrition, human milk also contains beneficial compounds [36], such as 2FL. In this study, we demonstrated that 2FL, a bioactive component in human milk, has protective effects against ICH. 2FL may have clinical implications for the treatment of ICH.

## 4. Materials and Methods

### 4.1. Animals

Adult male and time-pregnant Sprague-Dawley rats were purchased from BioLASCO, Taipei, Taiwan. The use of animals was approved by the Animal Research Committee of National Health Research Institutes of Taiwan (NHRI-IACUC106101-A). All animal experiments were carried out in accordance with the National Institutes of Health Guide for the Care and Use of Laboratory Animals (NIH Publications No. 8023, revised 1978).

### 4.2. Materials

2′-Fucosyllactose was provided by Advanced Protein Technologies Corp. (Suwon-si, Gyeonggi-do Province, Korea). Bovine serum albumin, chloral hydrate, fetal bovine serum, l-glutamate, paraformaldehyde, poly-d-lysine, hemin, and Triton X-100 were purchased from Sigma (St. Louis, MO, USA). Alexa Fluor 488 (secondary antibody), B27 supplement, Dulbecco’s modified Eagle’s medium, Neurobasal Medium, and trypsin were purchased from Invitrogen (Carlsbad, CA, USA). Anti-CD4 antibody was purchased from Proteintech (Rosemont, IL, USA). Anti-MAP2 was purchased from Millipore (Burlington, VT, USA). Anti-IBA1 antibody was purchased from Wako (Richmond, VA, USA).

### 4.3. Primary Rat Cortical Neuron (PCN) and Microglia Coculture

Primary cortical neuron (PCN) cultures were prepared from embryonic (E14–15) cortex tissues obtained from the fetuses of term-pregnant Sprague-Dawley rats. After removing the blood vessels and meninges, pooled cortices were trypsinized (0.05%; Invitrogen, Carlsbad, CA, USA) for 20 min at room temperature. After rinsing off trypsin with prewarmed Dulbecco’s modified Eagle’s medium (Invitrogen, Carlsbad, CA, USA), cells were dissociated by trituration, counted, and plated into 96-well (5.0 × 10^4^/well) cell culture plates precoated with poly-d-lysine (Sigma-Aldrich, St. Louis, MO, USA). The culture plating medium consisted of neurobasal medium supplemented with 2% heat-inactivated FBS, 0.5 mmol/L l-glutamine, 0.025 mM l-glutamate, and 2% B27 (Invitrogen, Carlsbad, CA, USA). Cultures were maintained at 37 °C in a humidified atmosphere of 5% CO_2_ and 95% air. The cultures were fed by exchanging 50% of medium with feeding medium (neurobasal medium), 0.5 mmol/L l-glutamine, and 2% B27 with antioxidant supplement on days in vitro (DIVs) 3 and 5. BV2 microglia were cultured separately, detached by 0.05% trypsin-ethylenediaminetetraacetic acid (EDTA, Invitrogen), and centrifuged at 100× *g* for 5 min. BV2 cells were resuspended in the feeding medium containing B27 supplement without antioxidants (−AO, from Invitrogen, Carlsbad, CA, USA). The density of surviving cells was counted using a trypan blue assay; cells were plated on the PCN plated wells at a concentration of 3.0 × 10^3^/well on DIV 7, as previously described [37]. The cocultures were fed with −AO medium on DIVs 7 and 10. On DIV 10, cultures were treated glutamate with 2FL or vehicle. At 48 h after drug treatment, cells were fixed with 4% paraformaldehyde (PFA, Sigma-Aldrich, St. Louis, MO, USA) for 1 h at room temperature.

### 4.4. Immunocytochemistry

After removing 4% PFA solution, cells were washed with PBS. Fixed cells were treated with a blocking solution (5% BSA and 0.1% Triton X-100 in PBS) for 1 h. The cells were incubated for 1 day at 4 °C with a mouse monoclonal antibody against MAP2 (1:500; Millipore, Billerica, MA, USA) and rabbit polyclonal antibody against IBA1 (1:500; Wako, Richmond, VA, USA), before rinsing three times in PBS. The bound primary antibody was visualized using AlexaFluor 488 goat anti-mouse or AlexFluoro 568 goat anti-rabbit secondary antibody (Invitrogen, Carlsbad, CA, USA). Images were acquired using a camera DS-Qi2 (Nikon, Tokyo, Japan) attached to a NIKON ECLIPSE Ti2 (Nikon, Tokyo, Japan) inverted microscope by blinded observers. The pixel density of MAP2-ir or IBA1-ir was analyzed using NIS Elements AR 5.11 Software (Nikon).

### 4.5. Surgery

Rats were housed in a 12 h dark (7 p.m. to 7 a.m.) and 12 h light (7 a.m. to 7 p.m.) cycle. Animals were anesthetized and placed in a stereotaxic frame. Type VII collagenase (0.5 U/μL × 1.0 μL, C-0773, Sigma Aldrich, St. Louis, MO, USA) was stereotactically injected into the right striatum (coordinates: 0.0 mm rostral and 3.0 mm lateral to bregma, 5.5 mm below the skull) at 0.4 μL/min over 5 min on day 0. Then, 2FL (400 mg/kg/day × 5 days) or vehicle was administered i.p. from days 1 to day 5. Animals were sacrificed on day 5 for histological and PCR analysis.

### 4.6. Locomotor Behavioral Measurement

Locomotion was measured on day 5 using an infrared activity monitor (Accuscan, Columbus, OH, USA). Rats were individually placed in a 3D infrared behavior chamber (42 × 42 × 21 cm) for 120 min. Six variables were measured: (i) vertical activity (VACTV, the total number of beam interruptions that occurred in the vertical sensors), (ii) total distance traveled (TOTDIST, the distance, in centimeters, traveled by the animals), (iii) vertical movement time (VTIME), (iv) horizontal activity (HACTV, the total number of beam interruptions that occurred in the horizontal sensors), (v) horizontal movement time (MOVTIME), and (vi) number of vertical movements (VMOVNO).

### 4.7. Immunohistochemistry

Animals were anesthetized and perfused transcardially with saline followed by 4% PFA in phosphate buffer (PB; 0.1 mol/L; pH 7.2); they were post-fixed for 18–20 h and then transferred to 20% sucrose in 0.1 M PB for at least 16 h. Serial sections of brains were cut at a 30 μm thickness using a cryostat (model: CM 3050 S; Leica, Heidelberg, Germany). Brain sections were rinsed in PB and were blocked with 4% bovine serum albumin (Sigma-Aldrich) with 0.3% Triton X-100 (Sigma-Aldrich) in 0.1 mM PB. Brain slices were then incubated with primary antibodies against CD4 (polyclonal 1:100, proteintech, Rosemont, USA) or IBA1 (monoclonal 1:100, Wako, Richmond, VA, USA) at 4 °C overnight. Sections were rinsed in 0.1 mM PB and incubated in Alexa Fluor 488 secondary antibody solution (1:500; Molecular Probes, Eugene, OR, USA). Control sections were incubated without the primary antibody. Brain sections were mounted on slides and coverslipped. Confocal analysis was performed using a Nikon D-ECLIPSE 80i microscope (Nikon Instruments, Inc., Tokyo, Japan) and EZ-C1 3.90 software (Nikon, Tokyo, Japan). The optical density of IBA1 or CD8 immunoreactivity was quantified in two consecutive brain sections with a visualized anterior commissure in each animal. Two photomicrographs were taken along the peri-lesioned region per brain slice; IBA1 or CD4 optical density was analyzed using NIS Elements AR 3.2 Software (Nikon) and was averaged in each brain for statistical analysis. All immunohistochemical measurements were performed by blinded observers.

### 4.8. Quantitative Reverse Transcription PCR (qRTPCR)

Striatal tissues from the lesioned and non-lesioned hemispheres were collected. Total RNAs were isolated using TRIzol Reagent (ThermoFisher, #15596-018, Waltham, MA, USA), and cDNAs were synthesized from 1 μg of total RNA by use of a RevertAid H Minus First-Strand cDNA Synthesis Kit (Thermo Scientific, #K1631, Waltham, MA, USA). cDNA levels for CD86, CD206, TGFβ, PERK, IRE1, CHOP, Sigmar1, BIP, ATF6, caspase3, actin, and GAPDH were determined using specific universal probe library primer-probe sets or gene-specific primers (Table 2). Samples were mixed with TaqMan Fast Advanced Master Mix (Life Technologies, #4444557, Carlsbad, CA, USA) or SYBR (Luminaris Color HiGreen Low ROX qPCR Master Mix; ThermoScientific, Waltham, MA, USA). Quantitative real-time PCR (qRT-PCR) was carried out using the QuantStudio™ 3 Real-Time PCR System (ThermoScientific, Waltham, MA, USA). The expression of the target genes was normalized relative to the endogenous reference genes (beta-actin and GAPDH averages) using a modified delta-delta-Ct algorithm. All experiments were carried out in duplicate.

### 4.9. Statistics

Data are presented as the mean ± SEM. An unpaired *t*-test or a one- or two-way ANOVA was used for statistical comparisons, with a significance level of *p* < 0.05. In the event of multiple comparisons, a post hoc Newman–Keuls test was performed.

## Figures and Tables

**Figure 1 ijms-22-09881-f001:**
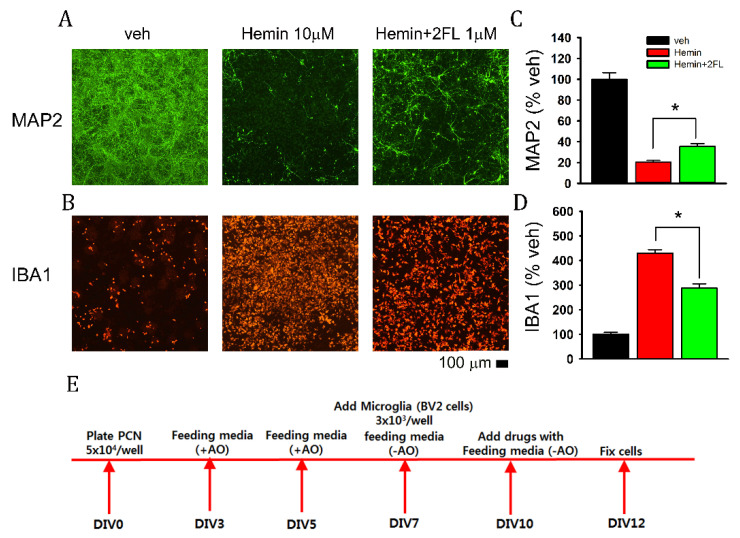
The neuroprotective effect of 2FL in a primary cortical neuron and BV2 microglia coculture. (**A**) Representative photomicrographs demonstrate that hemin reduced MAP2 immunoreactivity (-ir). Coadministration with 2FL antagonized the hemin-mediated loss of MAP-ir. (**B**) Hemin increased IBA1-ir. Coadministration with 2FL reduced IBA1-ir. (**C**,**D**) Pixel density of MAP2-ir or IBA1-ir was analyzed using NIS Elements AR 5.11 Software. (**C**) 2FL significantly antagonized hemin-mediated loss of MAP2-ir. (**D**) 2FL significantly attenuated hemin-induced IBA1 activity. (**E**) Timeline of experiment. * *p* < 0.05. Scale bar = 100 μm.

**Figure 2 ijms-22-09881-f002:**
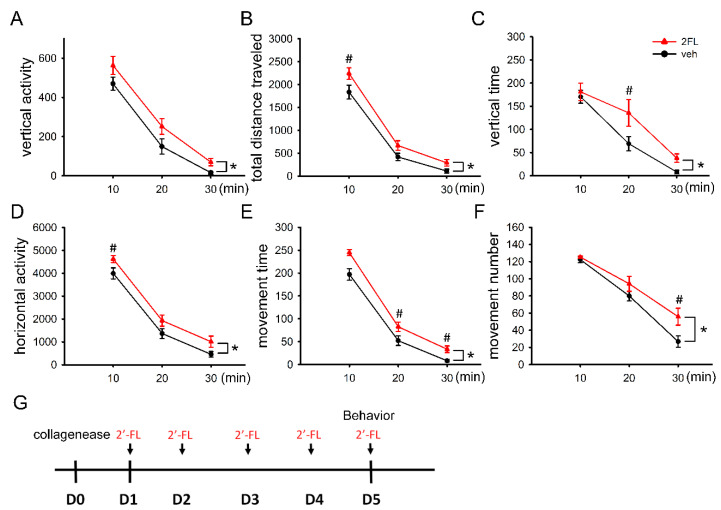
Post-treatment with 2FL improves neural functions in ICH rats. Adult rats were treated with type VII collagenase on D0 (**G**, timeline). 2FL or vehicle was given systemically daily for 5 days. Locomotor behavior was examined on D5. 2FL significantly improved (**A**) vertical activity, (**B**) total distance traveled, (**C**) vertical time, (**D**) horizontal activity, (**E**) movement time, and (**F**) movement number. *n* = 7 in each group. * *p* < 0.05, two-way ANOVA; ^#^ *p* < 0.05, post hoc NK test.

**Figure 3 ijms-22-09881-f003:**
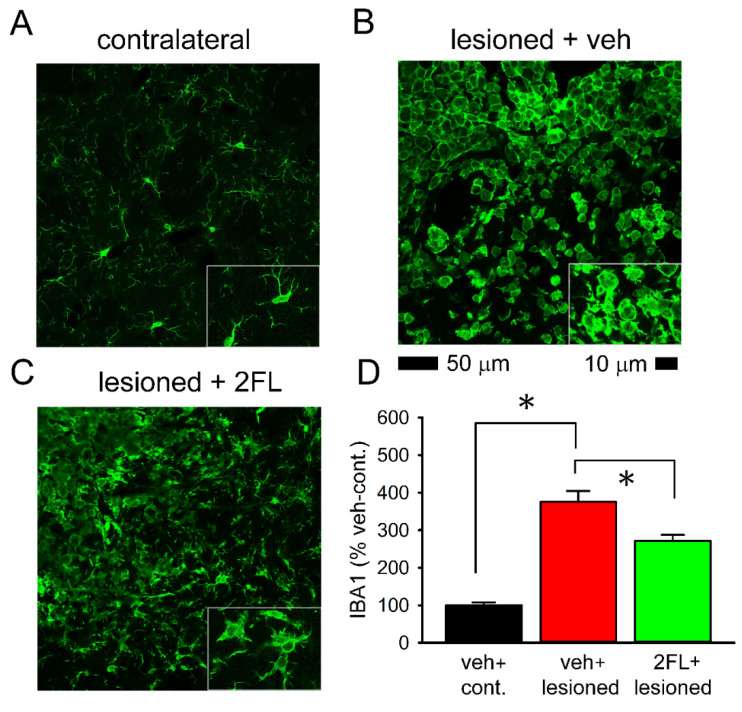
Early post-treatment 2FL reduced microglial activation in the lesioned striatum. Animals received vehicle or 2FL after collagenase injection from days 1 to 5. Animals were sacrificed on day 5. Enhanced IBA1 immunoreactivity with de-ramified morphology was found in the lesioned striatum (**B** vs. **A**). Treatment with 2FL reduced IBA immunoreactivity (**C** vs. **B**). (**D**) 2FL significantly reduced IBA1 immunoreactivity in the lesioned brain. * *p* < 0.05, one-way ANOVA + NK test.

**Figure 4 ijms-22-09881-f004:**
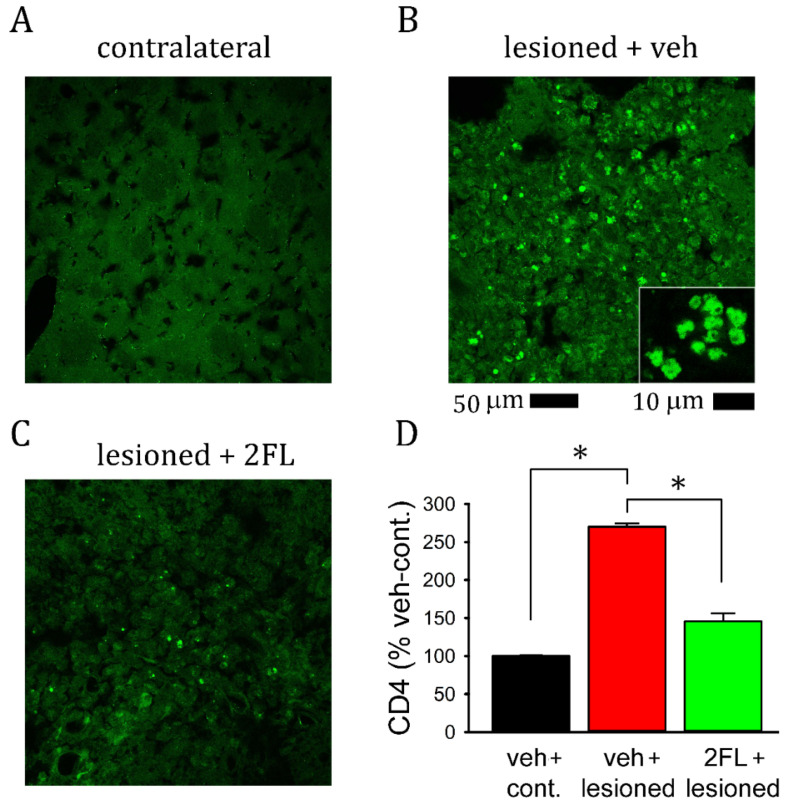
Post-treatment with 2FL reduced CD4^+^ lymphocyte infiltration into the lesioned brain. Representative photomicrographs demonstrate the CD4 immunoreactivity in (**A**) the non-lesioned striatum and (**B**) the lesioned striatum from a rat receiving vehicle at low (calibration = 50 μm) and high magnification (insert, calibration = 10 μm), as well as (**C**) the lesioned striatum from a rat receiving 2FL. (**D**) 2FL significantly reduced CD4-ir in the lesioned brain (* *p* < 0.001, one-way ANOVA).

**Figure 5 ijms-22-09881-f005:**
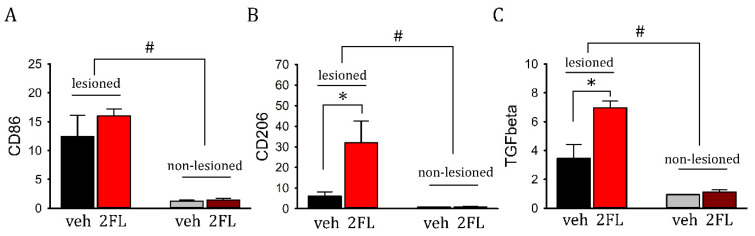
2FL upregulated the expression of M2 markers in ICH brains. ICH significantly increased the expression of (**A**) M1 marker CD86, and M2 markers (**B**) CD206 and (**C**) TGFβ (^#^ *p* = 0.005, two-way ANOVA). (**A**) 2FL significantly upregulated the expression of (**B**) CD206 and (**C**) TGFβ, but not (**A**) CD86, in the lesioned striatum (* *p* < 0.05, NK test).

**Figure 6 ijms-22-09881-f006:**
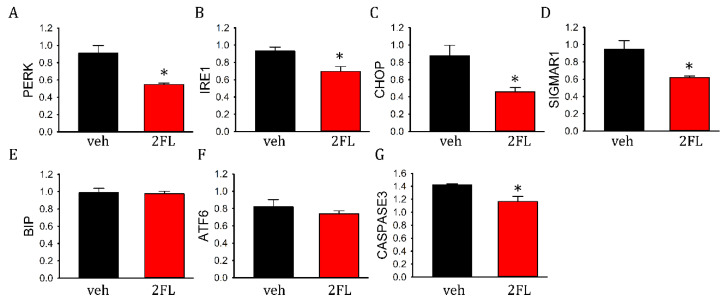
2FL suppressed the expression of ER stress and apoptotic markers. The expression of (**A**) PERK, (**B**) IRE1, (**C**) CHOP, (**D**) SigmaR1, and (**G**) caspase3 was significantly inhibited by 2FL in hemorrhagic brains. The expression of (**E**) BIP and (**F**) ATF6 was not altered by 2FL. * *p* < 0.05. ER, endoplasmic reticulum; PERK, protein kinase R-like ER kinase; IRE1, inositol-requiring enzyme 1; CHOP, CCAAT enhancer-binding protein homologous protein; SigmaR1, Sigma 1 receptor; ATF6, activating transcription factor 6.

**Table 1 ijms-22-09881-t001:** Significant differences in locomotor behaviors between ICH +veh and ICH +2FL.

	ICH Veh vs. ICH 2FL		
	F Value	Number of Animal	*p* Value
VACTV	5.527	14	0.024
TOTDIST	11.173	14	0.002
VTIME	6.676	14	0.014
HACTV	11.534	14	0.002
MOVTIME	22.421	14	<0.001
MOVNO	8.003	14	0.008

VACTV: vertical activity, TOTDIST: total distance traveled, VTIME: vertical time, HACTV: horizontal activity, MOVTIME: movement tine, MOVNO: movement number, 2FL: 2-fucosyllactose. *p*-value was determined by a two-way ANOVA + NK test.

**Table 2 ijms-22-09881-t002:** Oligonucleotide primers used for quantitative RT-PCR.

Gene	SYBR Green		TagMan
	Forward	Reverse	
CD86	TAGGGATAACCAGGCTCTAC	CGTGGGTGTCTTTTGCTGTA	
CD206	AGTTGGGTTCTCCTGTAGCCCAA	ACTACTACCTGAGCCCACACCTGCT	
TGFβ	GCTGAACCAAGGAGACGGAAT	CGGTTCATGTCATGGATGGTG	
PERK	GAAGTGGCAAGAGGAGATGG	GAGTGGCCAGTCTGTGCTTT	
IRE1	TCATCTGGCCTCTTCTCTCGGA	TTGAGTGAGTGGTTGGAGGC	
CHOP	ACCACCACACCTGAAAGCAG	AGCTGGACACTGTCTCAAAG	
Sigmar1	AAAGTGAGGTCTATTACCCAGGAG	TTTGGTCCCCACTCCACA	
Bip	TCGACTTGGGGACCACCTAT	GCCCTGATCGTTGGCTATGA	
ATF6	GGACCAGGTGGTGTCAGAG	GACAGCTCTGCGCTTTGGG	
Caspase3	GTGGAACTGACGATGATATGGC	CGCAAAGTGACTGGATGAACC	
β-Actin			Rn00667869_m1
GAPDH			Rn01775763_g1

## Data Availability

The data that support the findings of this study are available from the corresponding author upon reasonable request.

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
