# Peer review of "Human Milk Oligosaccharide 2′-Fucosyllactose Induces Neuroprotection from Intracerebral Hemorrhage Stroke"

_ijms, 2021, doi:10.3390/ijms22189881_

Round 1
Reviewer 1 Report
The authors present an experimental study to examine the neural protective actions of 2FL in cellular and rodent models of Intracerebral hemorrhage (ICH). The authors found that 2FL significantly mitigated ICH-induced microglia activation, CD4(+) lymphocyte infiltration, and ER stress in the injured brain and improved behavioral function (improving locomotor activity and movements in ICH rats). The authors suggest that 2FL is anti-inflammatory and neuroprotective in a cellular model of ICH. Therefore, the main finding of the study is that 2FL is neuroprotective against ICH injury. The authors suggest that 2FL may have clinical implications for the treatment of ICH. This is a robust study with an adequate design. Some comments are intended to improve the quality of the presentation:
1.Typographical errors (Intracerebral "hemorrhagic") in the Abstract should be corrected
2.Please change "major neurodegenerative disorder" by "major neurological disorder" in the Introduction.
3.Intracerebral hemorrhage (ICH) constitutes 10-15% of all acute stroke cases. To highlight the epidemiological importance of cerebrovascular diseases, it would be useful to mention in the Introduction the results of an epidemiologic study in Catalonia (Spain) on acute stroke (Rev Esp Cardiol 2007; 60; 573-580). In this study, the cumulative incidence of cerebrovascular diseases per 100,000 population was 218 (95% CI, 214-221) in men and 127 (95% CI, 125-128) in women.
4.The authors have certainly and adequately mentioned the clinical severity of brain hemorrhages associated with a high risk of death in adults. However, it would also be interesting to mention that there is a subgroup of intracerebral hemorrhages of small size of the lesion which manifests clinically as a lacunar syndrome (hemorrhagic lacunar stroke). This subgroup accounts for 7.4% of cerebral hemorrhages and is associated with a more favorable prognosis compared to the remaining group of intracerebral hemorrhages, usually without in-hospital deaths and symptom-free at discharge in 22.8% of cases, as described in a clinical series of this condition (see Cerebrovasc Dis 2000;10:229-234). The inclusion and comment on this reference is recommended.
5.It would be interesting to include a comment on the fact that this study opens new encouraging perspectives for the protection of the brain from acute hemorrhagic injury. A prospective randomized trial of 2FL treatment in humans may be justified.
Author Response
Point-to-point responses to the comments of reviewers
Reviewer #1:
The authors present an experimental study to examine the neural protective actions of 2FL in cellular and rodent models of Intracerebral hemorrhage (ICH). The authors found that 2FL significantly mitigated ICH-induced microglia activation, CD4(+) lymphocyte infiltration, and ER stress in the injured brain and improved behavioral function (improving locomotor activity and movements in ICH rats). The authors suggest that 2FL is anti-inflammatory and neuroprotective in a cellular model of ICH. Therefore, the main finding of the study is that 2FL is neuroprotective against ICH injury. The authors suggest that 2FL may have clinical implications for the treatment of ICH. This is a robust study with an adequate design. Some comments are intended to improve the quality of the presentation:
A: We would like to thank the reviewer 1 to consider our study is a robust study with an adequate design.
Point 1: Typographical errors (Intracerebral "hemorrhagic") in the Abstract should be corrected.
A: Thanks for your suggestion. We have revised typos.
Point 2: Please change "major neurodegenerative disorder" by "major neurological disorder" in the Introduction.
A:. We have made changes accordingly (1st paragraph, page 2).
Point 3: Intracerebral hemorrhage (ICH) constitutes 10-15% of all acute stroke cases. To highlight the epidemiological importance of cerebrovascular diseases, it would be useful to mention in the Introduction the results of an epidemiologic study in Catalonia (Spain) on acute stroke (Rev Esp Cardiol 2007; 60; 573-580). In this study, the cumulative incidence of cerebrovascular diseases per 100,000 population was 218 (95% CI, 214-221) in men and 127 (95% CI, 125-128) in women.
A: The epidemiological study and references were now included in the Introduction (1st paragraph, page 2).
4.The authors have certainly and adequately mentioned the clinical severity of brain hemorrhages associated with a high risk of death in adults. However, it would also be interesting to mention that there is a subgroup of intracerebral hemorrhages of small size of the lesion which manifests clinically as a lacunar syndrome (hemorrhagic lacunar stroke). This subgroup accounts for 7.4% of cerebral hemorrhages and is associated with a more favorable prognosis compared to the remaining group of intracerebral hemorrhages, usually without in-hospital deaths and symptom-free at discharge in 22.8% of cases, as described in a clinical series of this condition (see Cerebrovasc Dis 2000;10:229-234). The inclusion and comment on this reference is recommended.
A: We have included the reference of hemorrhagic lacunar stroke as suggested (Introduction).
5.It would be interesting to include a comment on the fact that this study opens new encouraging perspectives for the protection of the brain from acute hemorrhagic injury. A prospective randomized trial of 2FL treatment in humans may be justified.
A: Thanks for your suggestion. These information have been included in the Discussion (5th paragraph, page 9).
Reviewer 2 Report
The article entitled “Human milk oligosaccharide 2'-fucosyllactose induces neuroprotection in intracerebral hemorrhage stroke brain” nicely demonstrated the neuroprotective potential of 2FL in in vitro and in vivo models of ICH. In neuron and microglia co-culture, 2FL mitigated hemin-induced microglial activation and neurodegeneration. In rat model of ICH, 2FL improved locomotor activity parameters, and reduced microglial activation and CD4 T cells infiltration. However, there are many points that should be considered:
- In figure 1, the software used for measuring the pixel density of the images was not mentioned. Also, it is not necessary to assign images as A1, A2, A3…etc; it is enough to mark them as Fig1A, B, C ..etc.
- The effect of 2FL on the hemorrhagic volume “hematoma volume” was not investigated. If it was investigated and didn’t show any effect, the results should be mentioned even in the supplementary file. If it was not investigated, it should be mentioned in the limitation of the study. Improvement in locomotor activity denotes a reduction in the hemorrhagic volume.
- Page 7, paragraph 2.6.2: the method used for testing the expression of ER stress and apoptotic markers was not mentioned in the paragraph nor in the figure legend. It should be mentioned at least in one of them.
- In the discussion section, third paragraph, line 181: “2FL differently regulates the expression of M1 (proinflammatory)”. However, in the result section (line 134), 2FL didn’t change the expression of CD86. Therefore, this statement should be corrected.
- limitations of the study should be mentioned in the discussion section.
- The units should be consistent: in line 65 Hemin (10 μM) while in line 67 2FL (1uM).
- In lines 127 and 128, the stratum should be corrected to the striatum.
- Line 136, TGFb in the lesioned side (p<0.05, NK test). “In the lesioned striatum” is just repetition and should be removed.
- Line 208, 2Fl should be corrected to 2FL.
Author Response
Point-to-point responses to the comments of reviewers
Reviewer #2:
The article entitled “Human milk oligosaccharide 2'-fucosyllactose induces neuroprotection in intracerebral hemorrhage stroke brain” nicely demonstrated the neuroprotective potential of 2FL in in vitro and in vivo models of ICH. In neuron and microglia co-culture, 2FL mitigated hemin-induced microglial activation and neurodegeneration. In rat model of ICH, 2FL improved locomotor activity parameters, and reduced microglial activation and CD4 T cells infiltration. However, there are many points that should be considered:
- In figure 1, the software used for measuring the pixel density of the images was not mentioned. Also, it is not necessary to assign images as A1, A2, A3…etc; it is enough to mark them as Fig1A, B, C ..etc.
A: We have included the name and source of software used for measuring the pixel density in figure legend and Methods 4.4. The numbers (A1, A2, A3..) in Figure 1 were deleted.
- The effect of 2FL on the hemorrhagic volume “hematoma volume” was not investigated. If it was investigated and didn’t show any effect, the results should be mentioned even in the supplementary file. If it was not investigated, it should be mentioned in the limitation of the study. Improvement in locomotor activity denotes a reduction in the hemorrhagic volume.
A: We did not measure the hematoma volume. In our study, brain tissues were fixed in PFA and sectioned into 30 µm sections. Free-floating immunohistochemistry was performed in a 24 well plate. The hematoma or liquefied dead tissues at the core of injury were washed in the free-floating preparation. These information were now included in the limitation of study (5th paragraph, page 9).
- Page 7, paragraph 2.6.2: the method used for testing the expression of ER stress and apoptotic markers was not mentioned in the paragraph nor in the figure legend. It should be mentioned at least in one of them.
A: The expression of ER stress marker (PERK, IRE1, CHOP, Sigmar1, BIP, and ATF6) and apoptotic marker (Caspase-3) was examined by qRTPCR. These information were now included in Methods 4.8.
- In the discussion section, third paragraph, line 181: “2FL differently regulates the expression of M1 (proinflammatory)”. However, in the result section (line 134), 2FL didn’t change the expression of CD86. Therefore, this statement should be corrected.
A: We have now rephrased the sentences.
- limitations of the study should be mentioned in the discussion section.
A: We added the limitation of the study (5th paragraph, page 9).
- The units should be consistent: in line 65 Hemin (10 μM) while in line 67 2FL (1uM).
A: All units were now checked and corrected. .
- In lines 127 and 128, the stratumshould be corrected to the striatum.
A: The typos have been corrected.
- Line 136, TGFb in the lesioned side (p<0.05, NK test). “In the lesioned striatum” is just repetition and should be removed.
A: The repetitions were removed.
- Line 208, 2Flshould be corrected to 2FL.
A: We have made the changes accordingly.